# Evaluating N95 respirator designs: A mixed-methods pilot and feasibility study

Fatima Sheikh[1], Myrna Dolovich[2,3], Lisa Schwartz[1], Sarah Khan[4,5], Zeinab Hosseinidoust[6], Alison E. Fox-Robichaud[1,2,5]*

1 Department of Health Research Methods, Evidence and Impact, McMaster University, Hamilton, Ontario, Canada, 2 Department of Medicine, McMaster University, Hamilton, Ontario, Canada, 3 St. Joseph's Hospital, Firestone Institute for Respiratory Health, McMaster University, Hamilton, Ontario, Canada, 4 Department of Pediatrics, McMaster University, Hamilton, Ontario, Canada, 5 Hamilton Health Sciences, Hamilton, Ontario, Canada, 6 Department of Chemical Engineering, McMaster University, Hamilton, Ontario, Canada

* afoxrob@mcmaster.ca

## Abstract

### Background

Severe acute respiratory syndrome coronavirus 2 (SARS-CoV-2) has had a global impact, underscoring the importance of personal protective equipment (PPE). Use of N95s reduces the risk of airborne infection; however, in the absence of equitable designs, health care workers (HCWs) who do not fit the average White male head and face are at an increased risk of airborne infectious diseases.

### Objectives

Primary: Feasibility of a mixed-method study, with a sample size of 100, 50% of participants self-identifying as non-White and having at least one characteristic of interest. Secondary: (1) Generate quantitative evidence on N95 fit using a PortaCount fit test, (2) describe participant-reported feelings on fit and breathability, and (3) evaluate the impacts of the pandemic on a HCW's physical and mental well-being.

### Methods

This was a mixed-method prospective pilot and feasibility study. Quantitative fit was assessed using a TSI PortaCount test and measurements of bizygomatic breadth and Menton-Sellion length. A survey was administered to collect sociodemographic information, HCWs' assessment of N95 fit, comfort, and the impact of PPE-related challenges on well-being.

### Results

This study was limited by a small sample size, as COVID-19 pandemic restrictions prevented adequate recruitment to detect differences between groups. We describe

**Data availability statement:** All relevant data are within the manuscript and its Supporting Information files.

**Funding:** This study was funded by the Canadian Institute for Health Research (CIHR) (202005VR4-447804-CCF) for one year beginning June 1st, 2020. FS was also supported by an Ontario Graduate Fellowship Award during the conduct of this study. The funders had no role in study design, data collection and analysis, decision to publish, or preparation of the manuscript.

**Competing interests:** The authors have declared that no competing interests exist.

key findings that should inform analyses of the impact of gender and ethnicity on N95 respirator fit. Following a study amendment to increase eligible sites, 37 of the 41 (90.2%) approached HCWs consented to participate. Compared to other HCWs, non-White females had the lowest mean fit factor. Differences in Menton-Sellion length and bizygomatic breadth were observed between males, females, and White and non-White HCWs. Most HCWs reported physical discomfort and negative impacts on their psychological well-being.

## Conclusions

We identified gender and ethnicity as key factors in the fit of N95s. Differences in gender, ethnicity, and anthropometric measures must be considered in respirator designs.

---

### Introduction

Coronavirus disease (COVID-19) has had a global impact, with more than 770 million confirmed cases and 6.9 million deaths reported worldwide [1]. Severe acute respiratory syndrome coronavirus 2 (SARS-CoV-2), the virus that causes COVID-19, can be transmitted by respiratory droplets and aerosols [2,3]. In healthcare settings, SARS-CoV-2 can be aerosolized during aerosol-generating procedures (AGP); aerosolized viral particles less than 5μm in size can pass through the pores of surgical masks. N95 respirators have been the recommended PPE for HCWs involved in caring for patients requiring AGPs throughout the pandemic [4,5]. SARS-CoV-2 and other airborne pathogens are an ongoing concern in healthcare settings.

N95 respirators offer the greatest protection and are essential for protection against airborne infectious agents [6–8]. According to the National Institute for Occupational Safety and Health (NIOSH), the national body that regulates N95 standards in the United States (US) and Canada, N95s must have at least 95% filtration efficiency to prevent inhalation of airborne particles less than 0.3μm in size and be closely fitted to the face [9]. The majority of current PPE, including masks and respirators, have been designed based on the anthropometric facial measurements of average men in the US and Europe [10,11]. The reliance on historical anthropometric data poses challenges for those who do not fit these criteria, including women [12–15]. In Canada, women represent 82% of HCWs [16]. With the vast majority of PPE designed for the average male head and face, a sizeable portion of the workforce may be wearing poorly fitting PPE for long periods. The inability to access appropriately fitting protection not only increases the risk of COVID-19, but can lead to physical and psychological discomfort [17].

This bias in design affects more than women. An international study of facial morphology found significant differences in nose height and width between Caucasian North Americans compared to Asian and Black ethnic groups, and significantly greater bizygomatic width in Caucasian men and Asians, among other differences [18]. Women and ethnic men who do not meet these standards are challenged to find well-fitted and comfortable N95s. In the absence of a tight fit or an appropriate seal,

N95s may not provide the required protection, putting HCWs at risk of contracting airborne pathogens [11]. Despite international research [14,19–22], no comparable studies have been conducted in Canada, where the healthcare workforce is increasingly diverse, underscoring the need for locally relevant evidence to inform respirator design and testing.

The purpose of this mixed-methods study was to [1] assess the feasibility of examining the fit of N95s in a diverse population of Canadian HCWs; [2] examine the outcomes of quantitative fit testing; and [3] describe qualitative measures of HCW-reported perceptions of N95 fit, breathability and comfort, and the impacts of PPE-related challenges on the physical and mental well-being of HCWs.

## Methods

### Study design

We conducted a mixed-method prospective pilot feasibility study designed to assess the qualitative and quantitative fit, comfort, and breathability of N95s in a diverse population of HCWs. A convergent parallel design was used in which qualitative and quantitative data were collected in parallel, analyzed separately, and then merged for interpretation [23,24]. Incorporation of the results of the quantitative fit test with the qualitative data of the survey occurred in the interpretation phase. The objectives, outcome measures and analysisof this study are shown in S1 Table.

### Quantitative study methods

To assess the fit of an N95, a trained professional performed a quantitative fit test, using the TSI 8030 PortaCount Respirator Fit Tester, according to the NIOSH standard testing procedures [25]. The quantitative fit test consisted of seven exercises designed to assess the amount of leakage around the face seal. The results of the quantitative test were recorded for each of the following exercises: normal breathing (performed at the start and end), deep breathing, turning the head side-to-side, nodding the head up and down, speaking out loud, and bending over. The overall fit factor, a ratio of aerosol concentration outside and inside the respirator, was also recorded [26,27]. A minimum overall fit factor of 100 was necessary to pass. If the individual failed the test with the first respirator, the fit test was stopped, reported as an unsuccessful fit, and repeated at least once more prior to testing with a different N95 respirator. The following respirators were available: 3M 1870+, 1860, 1860S, 1804S, and the Honeywell DC 365. Healthcare workers previously fitted to an N95 were offered the same respirator to start.

The bizygomatic breadth, defined as the maximum horizontal breadth of the face, and the Menton-Sellion length, defined as the distance between the Menton and Sellion landmarks, were measured using sliding calipers (VWR Model: 12777−830). Both measures were recorded in millimeters for each participant, according to NIOSH testing procedures [28]. These facial measurements were mapped to the NIOSH bivariate panel [28] corresponding to a range of face widths and lengths.

### Qualitative study methods

To determine the diversity of experiences with wearing N95s, we conducted a qualitative description study to explore the perspective of healthcare workers [29,30]. We designed a paper-based survey which was administered to frontline HCWs who underwent the quantitative fit test (S1 File). Prior to administering the survey, we assessed the suitability of the questions, readability, and overall clarity with a sample of critical care HCWs (n = 3). Minor changes were made to improve the clarity of questions. No questions were added or removed, and no other significant changes were made.

### Recruitment and participants

Potential participants included HCWs, defined as a healthcare provider (e.g., physician, nurse, respiratory therapist, etc.) or any patient-facing staff, at one of the three Hamilton Health Sciences (HHS) hospitals in Hamilton, Ontario, Canada.

Convenience and purposive sampling strategies were used to recruit participants who were readily accessible, willing to participate, and would facilitate an analysis of the effects of gender and ethnicity on the fit of N95s [31]. Participants were restricted to HCWs who worked on-site during the study period due to pandemic-related restrictions.

**Inclusion criteria**

- 18 years of age or older

- Healthcare worker or patient-facing staff at HHS

- Informed consent to participate in the fit test and complete the survey.

And 50% of the sample must also meet at least one of the following conditions:

- Self-identity as being non-White

- Have at least one of the following characteristics: religious head covering (e.g., Hijab, Turban), glasses, and/or facial hair (e.g., beard and/or mustache)

- Identify as female

To ensure adequate representation of participants from diverse backgrounds, we established an a priori requirement that at least 50% of participants be non-White. This criterion was based on prior evidence that respirator fit may vary by ethnicity and aimed to avoid overrepresentation of White participants, thereby supporting a more inclusive and equity-focused analysis.

*Exclusion criteria*

- Unable to safely complete a PortaCount fit test, including, for example, pre-existing respiratory conditions or other medical or safety concerns.

## Analysis

**Co-primary feasibility outcomes.** The co-primary outcomes are described in detail in S1 Table.

**Secondary outcomes**

(1) Quantitative Fit Test Results. The results of the quantitative fit test are described with descriptive statistics. Specifically, the results for each of the seven exercises and the overall fit factor are reported as mean±standard deviation (SD). These results are presented in aggregate, by gender, and by ethnicity. Data were reported by male and female, as no other genders were reported, and as White and Non-White due to limited sample size across each individual ethnicity. All data analysis was performed using *SPSS Statistics Version 27* (IBM, Chicago, IL). Due to the small sample size and risk of detecting a significant difference between males and females or the various ethnic groups, when, in fact, there was no difference (i.e., Type II error), no inferential statistics were run [32]. Instead, descriptive statistics were presented to characterize patterns in the data, complemented by qualitative findings to provide context.

(2) Measure of N95 Fit; and [3] Impacts related to PPE on physical and mental health. Domains evaluated in the survey using Likert scales, are summarized using medians and frequencies, where applicable [33]. Qualitative description was used to guide the reporting and analysis of the open-ended survey results, to ensure that the reported data directly reflect what the HCWs said, and how it was said [29]. HCW-reported data on N95s were narratively summarized, and reported in tables, organized by N95 fit, comfort, and breathability and their impacts on the physical and mental health of the HCWs.

To represent the convergent integration of both datasets, these data were summarized in the form of a "statistics-by-theme joint display" [34], where de-identified quotes are reported along with relevant numeric results and

sociodemographic information collected in the two phases of the study. Specifically, the integration of quantitative and qualitative findings was conducted through an iterative process, in which authors independently reviewed the quantitative and qualitative survey results and then collaboratively organized the findings into a joint display to highlight points of convergence and divergence. This approach ensured that both data sources were represented and that the joint display accurately reflected the descriptive nature of the qualitative findings.

### Ethics

Ethics approval was obtained from the Hamilton Integrated Research Ethics Board (project # 12776). Recruitment began on January 4th, 2022 and ended on May 6th, 2022. Participants provided written informed consent.

## Results

### Feasibility results

Feasibility results were collected up to May 6th, 2022. The flow of potentially eligible participants, beginning with the total number of fit tests conducted between January 4th and May 6th at Hamilton General Hospital (HGH), and between March 14th and May 6th, at Juravinski Hospital (JH) and McMaster Children's Hospital (MCH), are shown in Fig 1.

The results of the primary objectives, designed to assess the feasibility of conducting this study, are summarized in Table 1. Overall, study recruitment was well below the target (100 HCWs), with only 36 of the target 100 (36%) HCWs enrolled in the study. The primary reasons for low enrollment were the increased need for fit tests, the limited amount of time HCWs had, and for the first two months of the study, the limited number of days the fit test clinic was run. There were some additional challenges with equipment failure and the time required to have it repaired. Study recruitment increased slightly following the addition of two study sites (S2 File); however, challenges persisted, and the recruitment rate was low with only 36 of the 653 (5.5%) potentially eligible HCWs recruited. No other reasons, beyond time, were reported by HCWs or observed by the study team.

In the sample of HCWs approached for inclusion in the study, 37 of the 41 (90.2%) HCWs consented to participate in the study and 36 of the 41 (97.3%) were successfully fitted via the quantitative method. Additionally, 23 of the 36 (63.9%) included HCWs were fitted on the first try, while the remaining 13 required more than one fit test. One HCW was could not be fitted using the quantitative method due to equipment failure.

Among the 654 fit tests completed between January 4th and May 2nd, we were still unable to approach 612 potentially eligible HCWs due to time constraints or, on occasion, due to equipment failure. The research team felt it was important to ensure that the research did not "compromise the public health response to [the] outbreak or the provision of clinical care." [35] As a result, HCWs were approached for study participation if [1] there was no line-up at the fit test clinic, [2] the HCW did not express time constraints or the need to return to clinical duty, and [3] the addition of one more person to the room, while other fit tests were ongoing, did not exceed the capacity limit. A summary of the feasibility results are displayed in Table 1.

### Quantitative fit test results

Thirty-six HCWs were recruited and completed the quantitative fit test and survey (Fig 1). The characteristics of the included participants are reported in Table 2.

The overall mean fit factor for all HCWs was 173 (SD = 29.96), while male HCWs had a mean fit factor of 184 (SD = 28.0) compared to female HCWs, who on average had a fit factor of 170 (SD = 30.0). White HCWs had a fit factor of 178 (SD = 30.0), approximately 11 points higher than their non-White counterparts. Both the mean Menton-Sellion length, 122.87 mm (SD = 8.56) vs. 111.4 mm (SD = 9.37), and the mean bizygomatic breadth, 121.93 mm (SD = 8.4) vs. 109.26 mm (SD = 10.17), were longer in males compared to females. White HCWs had a slightly longer Menton-Sellion length, and non-White HCWs had a higher bizygomatic breadth value, compared to non-White and White HCWs, respectively.

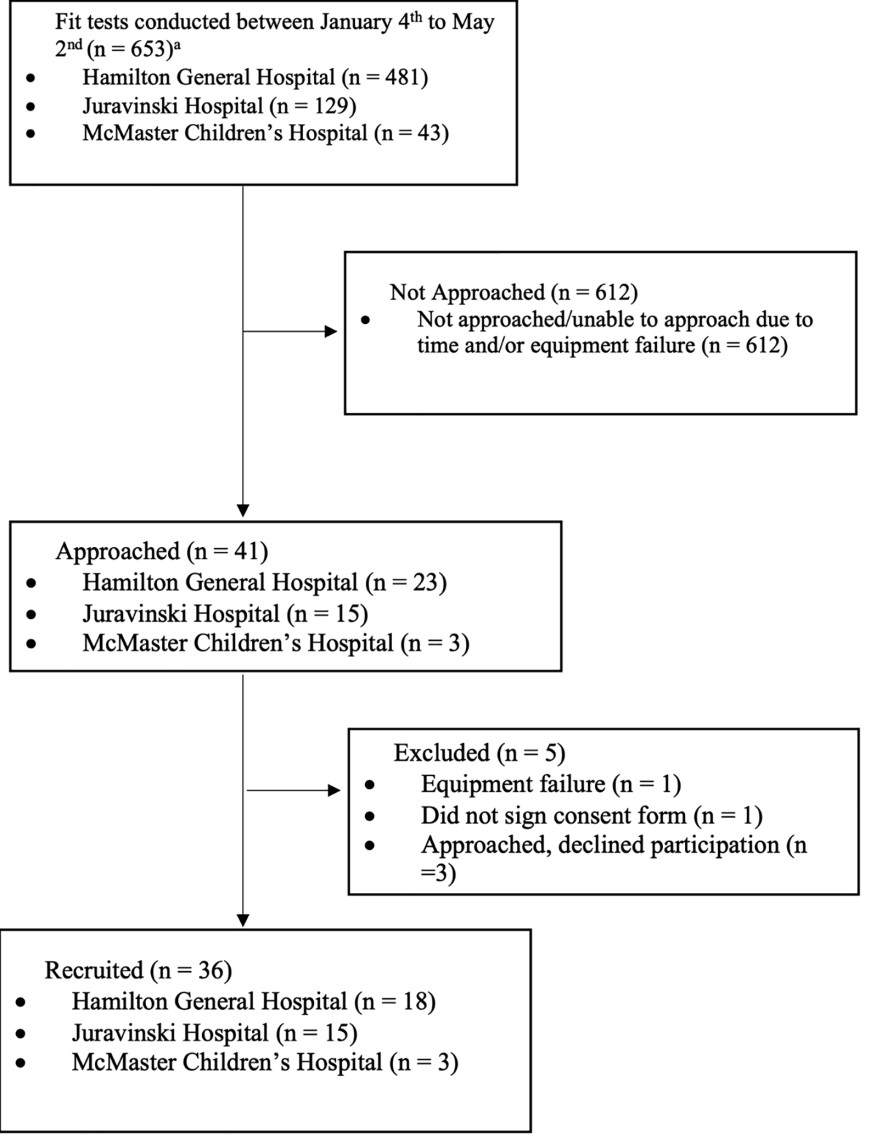

**Fig 1. Study Flow Diagram.** [a]Recruitment began at Juravinski and McMaster Children's Hospital on March 10th. The total number of fit tests completed at each of these sites prior to March 10th is not reported.

To assess the combined effects of gender and ethnicity, the results of the quantitative fit test from 35 HCWs are summarized by gender and ethnicity in Table 3. One HCW was excluded from this analysis as they were fitted to a P100, the results of which were out of range. White males had a mean fit factor of 200 (SD = 0), the maximum score attainable, and non-White males had a mean fit factor of 175 (SD = 31.0), compared to White and non-White females who had mean fit factors of 175 (SD = 32.0) and 165 (SD = 30.0), respectively.

The Menton-Sellion length was longer in White males compared to non-White males, and the bizygomatic breadth showed the opposite trend. Non-White female HCWs had the shortest Menton-Sellion length and bizygomatic breadth, with mean values of 108.99 mm (11.31) and 107.82 mm (9.12), compared to White female and male HCWs. When these values were assigned to the NIOSH bivariate panel, 27 of the 36 (75%) HCWs were out of range, 4 of the 36 (11.1%)

**Table 1. Summary of the Study Feasibility Outcomes.**

| Outcome | Outcome Description | Proportion (%) |
|---|---|---|
| Recruitment | A sample size of 100 HCWs* recruited within 4 months and 50% meet one of the following criteria:<br>• Self-identify as non-white<br>• Have one or more of the following characteristics: religious head covering, glasses, and/or facial hair<br>• Identify as female | 36/100 (36%)<br>Proportion of eligible HCWs: 36/653 (5.5%)<br>Characteristics of included participants are reported in Table 3. |
| Consent | Consent rate of ≥ 80% in approached HCWs. | 37/41 (90.2%) |
| Fit Test | Successful completion of a PortaCount fit test or partial completion with a reason for why the test was ended. | 36/37 (97.3%)<br>Proportion of successful fit tests on the first attempt:<br>23/37 (62.2%) |
| Survey | Completion of the survey, defined as at least 80% of the questions have been fully responded to. | 36/36 (100%) |

*HCW* Healthcare workers.

*100 HCWs in four months was selected as our feasibility target based on the number of people fit tested on average in our clinics, the duration of a full study, and in consultation with our multidisciplinary team.

**Table 2. Characteristics of Included Participants (n = 36).**

| Variable | N (%) |
|---|---|
| **Age,** (SD) | 31.97 (10.8) |
| **Gender** | |
| Male | 6/36 (16.7) |
| Female | 30/36 (83.3) |
| **Ethnicity** | |
| White (Caucasian) | 19 (52.8) |
| Black | 2 (5.6) |
| Chinese | 3 (8.3) |
| Filipino | 4 (11.1) |
| Latin American | 1 (2.8) |
| Polish* | 1 (2.8) |
| South Asian | 1 (2.8) |
| Southeast Asian | 1 (2.8) |
| Vietnamese* | 1 (2.8) |
| Mixed ethnicity | 2 (5.6) |
| **Physical Characteristics†** | |
| Glasses | 16 (44.4) |

*SD* Standard deviation.

*These ethnic categories were not included in the survey. The participants selected "prefer to self-identify". Responses are displayed as listed by participants.

†The fit test protocol requires individuals to be clean-shaven. For safety reasons, and in the context of COVID-19, we were unable to include HCWs with facial hair (even for research purposes). No individuals with a head covering were recruited.

**Table 3. Summary of Quantitative Fit Test Results (n = 35).**

| Variable | Male (n = 6) | | Female (n = 29)* | |
| --- | --- | --- | --- | --- |
| | Non-White | White | Non-White | White |
| **Number of fit tests,** Mean (SD) | 2.25 (1.5) | 1 (0) | 1.62 (0.9) | 1.65 (1.1) |
| **Fit Factor,** Mean (SD) | 175 (31) | 200 (0) | 165 (30) | 175 (32) |
| Normal breathing | 200 (0) | 200 (0) | 185 (29) | 199 (5) |
| Deep breathing | 200 (0) | 200 (0) | 189 (26) | 189 (33) |
| Head side to side | 200 (0) | 200 (0) | 174 (49) | 180 (42) |
| Head up and down | 199 (1.5) | 200 (0) | 167 (55) | 161 (60) |
| Talking out loud | 168 (52) | 200 (0) | 196 (16) | 187 (30) |
| Bending over | 157 (66) | 200 (0) | 174 (52) | 180 (36) |
| Normal breathing | 173 (53) | 200 (0) | 168 (51) | 188 (23) |
| **Anthropometric Measures,** Mean (SD) | | | | |
| Menton-Sellion length | 122.07 (9.6) | 124.46 (9.0) | 108.88 (11.3) | 113.32 (7.3) |
| Bizygomatic breadth | 126.46 (4.7) | 112.87 (6.4) | 107.82 (9.1) | 110.36 (11.0) |

*SD* Standard deviation. *mm* Millimeters.

*All 36 included HCWs were fit tested; however, one female participant was excluded from the quantitative analysis because they were fitted to a P100, the results of which are out of range, relative to the N95 respirator fit test. Among the 35 HCWs included, 27 (75%) were fitted to a 3M 1870 + , 4 (11.2%) were fitted to a Honeywell DC 365, 3 (8.3%) were fitted to a 3M 1804s, and 1 (2.8%) was fitted to the 3M 1860s.

were assigned to panel 6, 3 (8.3%) were assigned to panel 3, 1 (2.8%) was assigned to panel 4, and 1 (2.8%) was assigned to panel 1 (Fig 2). This finding suggests that current respirator design and testing standards may not adequately capture the facial anatomy of a diverse healthcare workforce. The raw data to complete the quantitative analyses is found in S2 Table.

## Qualitative survey results

Thirty-six (100%) HCWs completed the survey and responded to at least 80% of the questions. On a scale of one to ten, where one is poor and ten is excellent, HCWs, on average, reported their experiences with N95s as a 6. There were no differences in reported experiences between White and non-White HCWs; however, male HCWs had a higher average score of 6.5, compared to female HCWs. Twenty-eight (77.8%) HCWs reported experiencing physical discomfort, 23 (63.9%) experienced pressure/pain, 14 (38.9%) experienced headaches, and 12 (33.3%) reported experiencing itching wearing an N95 respirator. Less frequent concerns included dizziness, reported by 8 (22.2%) HCWs, and nausea, reported by 2 (5.6%) HCWs.

A summary of quantitative and qualitative measures of N95 fit, comfort, and breathability, grouped by gender and ethnicity, are reported in Tables 4 and 5, respectively. On a five-point Likert scale, where 0 is poor and 4 is excellent or very good, both male and female HCWs had a median value of 3, corresponding to an agreement with the statement "N95 respirators fit me well". While male HCWs reported the comfort of N95 respirators as a 2, female HCWs reported the comfort of N95 respirators as a 1 out of 5. This was reflected in the open responses, with one female HCW stating that "With the 1860s mask, it was extremely uncomfortable, difficult to breathe in. I would say it fit fine, but everything else was not great." In terms of breathability, male HCWs reported an average score of 3, compared to females who reported an average score of 2. In addition to the perceived fit of N95s, females reported lower measures of N95 comfort and breathability.

When comparing the experiences of wearing N95s between White and non-White HCWs, the median scores for fit and breathability were the same. HCWs in each of these groups reported similar concerns, including respirators being too tight and causing red marks on their face when worn for prolonged periods of time. In terms of comfort, the median score in White HCWs was 1 compared to a median score of 2 in non-White HCWs. However, both groups reported feelings of

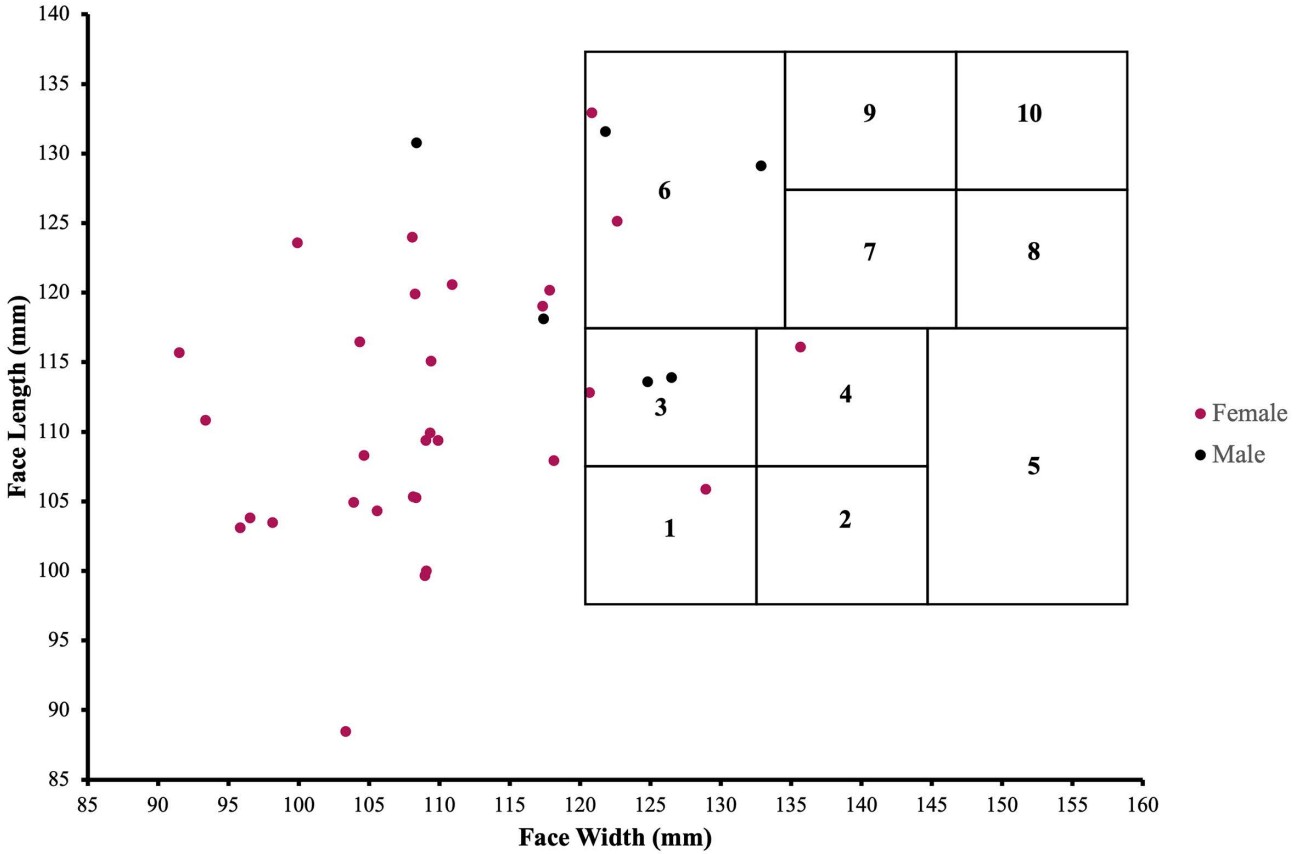

**Fig 2. Facial Anthropometric Measurements.** The facial measurements of the 36 included HCWs are displayed alongside the NIOSH bivariate panel. These panels correspond to a range of face widths (120.5–158.5 mm) and lengths (98.5–138.5 mm) and recommend number of participants to include from each panel when testing new respirators. Twenty-seven of the thirty-six (75%) included HCWs in this study were out of range, demonstrating differences in this sample and the testing standards.

Table 4. Summary of Survey Results, Reported by Gender.

| Domain | Median* (IQR) | | Domain descriptions & illustrative examples | |
|---|---|---|---|---|
| | **Male** | **Female** | **Male (n = 6)** | **Female (n = 29)** |
| Fit | 3 (1) | 3 (1) | "I use the 1860 N95 and have issues with the nose clamp." "Nose pain – bridge of nose" "Prolonged use was uncomfortable" | "Hard to get proper fit, some slide & move on face. Drawing blood – required to bend head to chest, lots of movement." "Uncomfortable after long periods of time. Difficult to get a comfortable seal across the nose." "With the 1860s mask it was extremely uncomfortable, difficult to breath in. I would say it fit fine, but everything else was not great." |
| Comfort | 2 (1) | 1 (1) | | |
| Breathability | 3 (1) | 2 (2) | | |

*IQR* Interquartile range.

*0 = Strongly Disagree, 1 = Disagree, 2 = Neutral, 3 = Agree, 4 = Strongly Agree to the statements N95s fit me/are comfortable/are breathable.

tightness and negative experiences exacerbated by wearing glasses or unique facial characteristics. For example, a HCW who self-identified as White stated "Find it challenging to wear with my glasses. Find my mask a bit awkward on my face. Breathable but give me headaches when I wear all day." Similarly, a HCW who self-identified as Vietnamese reported that it was "difficult finding appropriately sized masks due to small nasal bridge."

**Table 5. Summary of Survey Results, Reported by Ethnicity.**

| Domain | Median* (IQR) | | Domain descriptions & illustrative examples | |
|---|---|---|---|---|
| | White | Non-White | White (n = 19) | Non-White (n = 17) |
| Fit | 3 (1) | 3 (1) | "Fit tightly to point where causes facial markings, redness + bruising. Especially at bridge of nose." | "They feel tight but move around, not well formed to nose and chin." |
| Comfort | 1 (1) | 2 (1) | | |
| Breathability | 2 (2) | 2 (2) | "Find it challenging to wear with my glasses. Find my mask a bit awkward on my face. Breath-able but gives me headaches when I wear all day (but I am generally prone to headaches)." | "Difficult finding appropriately sized masks due to small nasal bridge." "Tight and quite uncomfortable." |

*IQR* Interquartile range.

*0 = Strongly Disagree, 1 = Disagree, 2 = Neutral, 3 = Agree, 4 = Strongly Agree to the statements N95s fit me/are comfortable/are breathable.

Additional qualitative results, including the impact of N95s on the mental health of HCWs, are summarized in S3 File. These direct excerpts illustrate the negative impacts of existing N95s on the mental health of healthcare workers, and, more importantly, the breadth of these impacts, including their ability to communicate with patients, feel safe while at work, and their overall comfort.

## Discussion

This study aimed to explore the fit, comfort, and breathability of N95 respirators in a diverse population of HCWs, and the first, to our knowledge, to compare fit testing outcomes with qualitative descriptions from the perspective of front-line HCWs during the peak of the COVID-19 pandemic. Although we were unable to approach the majority of potentially eligible HCWs who came to the fit test clinic, due to the increased need for fit tests and limited time, all HCWs with whom the project was discussed were eager to participate, highlighting the importance of this study and reflected in the high consent rate.

In addition to the primary feasibility outcomes, we evaluated the results of the quantitative N95 fit test and the survey data of the 36 included HCWs. When comparing the results between male and female HCWs, we found that, on average, male HCWs had a higher fit factor compared to female HCWs. In a study that examined the fit factor in a group of HCWs, Wardhan et al. found that women had higher fit failure rates, defined as a fit factor less than 100, compared to men [36]. These findings are consistent with studies by McMahon et al. [15] and Lee et al. [37], which demonstrated a 10% difference in fit failure rate between males and females, with females having a higher fit failure rate. Consistent with previous studies comparing anthropometric measures between males and females, we also identified differences in the key measures used to inform the design of N95s, between males and females.

When stratified by ethnicity, White HCWs had a higher fit factor compared to non-White HCWs, and both White and non-White males had higher fit factors, compared to their female counterparts. During the first wave of the COVID-19 pandemic, Green et al. analyzed the outcomes of fit tests across National Health Service hospitals in the United Kingdom. In addition to demonstrating differences in fit failure rates between men and women, they found that Black, Asian, and Minority Ethnic (BAME) HCWs had significantly higher fit failure rates compared to non-BAME HCWs [38]. Similarly, Chopra et al. found that females and BAME participants had lower fit factor scores and fit test pass rates, which was attributed to differences in facial features. Specifically, they identified 14 standardized anthropometric measures that were significantly smaller for females. Despite limited disaggregated data on facial measurements of BAME individuals, they also reported differences in facial geometry, face size and nose measurements between Asian, Black, and Caucasian individuals [21]. These differences in fit failure rates between men, women, and various ethnic groups can be explained by differences in head and face anthropometrics, as these studies have demonstrated significant differences between genders [12,13] and ethnicities [18,21,39].

There is a growing body of evidence on how gender and ethnicity affect respirator fit. Several studies report higher fit test failure rates among women and ethnic minority groups [14,19–21,40,41], while others attribute fit primarily to individual facial dimensions and the design of the respirator [12,22,42,43]. Our findings contribute to this literature by combining quantitative fit testing with descriptive qualitative insights from HCWs, who described discomfort, skin irritation, and psychological stress associated with respirator use – issues frequently noted by female and racialized participants. Ultimately, these differences highlight the importance of population-specific studies, as respirator fit reflects both individual and group-level anthropometric differences. To our knowledge, none of these studies have been conducted in Canada, where the healthcare workforce is increasingly diverse [44,45]; thus, locally gathered evidence is essential to guide respirator designs and fit testing protocols.

This study was limited by the number of HCWs we were able to recruit and the shortened study period and the absence of participants with religious head coverings or facial hair. The small sample size, specifically in the quantitative arm of our study, limited our analyses and our ability to run inferential statistics and report statistical differences. There may also have been differences in HCW characteristics and N95 fit testing outcomes between HCWs we were able to approach and HCWs we were unable to approach, which may limit the generalizability of the study results. Recruitment challenges were the result of the COVID-19 pandemic and persisted even after the study amendment was submitted. Despite these limitations, our study has several strengths: (1) this study of N95s in a diverse sample of HCWs, particularly in the absence of recognized Canadian standards by hospitals and limited Canadian respirator designs, is timely (2) used rigorous and objective methodology for conducting fit tests, and (3) included the collection of quantitative and qualitative data on N95 fit, comfort, and breathability.

In summary, we identified differences in the outcomes of the quantitative fit test and perceived measure of N95 and surgical mask fit between males, females, and various ethnic groups. Women make up approximately 82% of the current healthcare workforce, many of whom identify as a visible minority. Although recruitment challenges in this study were largely attributable to COVID-19 surge conditions, the disparities observed by gender and ethnicity are consistent with prior work and therefore likely to persist beyond the pandemic. The challenges around N95s and other PPE, including the negative impacts to physical and psychological well-being, will persist in the absence of equitable designs. Future studies, to assess the fit and comfort of current respirators, are necessary to inform evidence-based testing and new national standards for N95s used in healthcare settings. Specifically, future studies should (1) employ strategies for recruiting a truly diverse sample of HCWs, (2) include additional anthropometric measures that better account for face and head shape, and (3) explore factors such as occupation and duration of wear that may contribute to the fit and comfort of N95 respirators worn over the HCW working shift of 10–12 hours.

## Conclusion

This study was conducted during the COVID-19 pandemic, when restrictions on in-person research limited recruitment and resulted in a small sample size. While this precluded statistical comparisons between groups, the preliminary findings provide important insights into gender- and ethnicity-related variation in respirator fit among Canadian healthcare workers. The COVID-19 pandemic has highlighted and exacerbated concerns around the fit, comfort, and breathability of PPE, and specifically N95 respirators. The results of this pilot study highlight the differences in fit between males and females and HCWs of different ethnic groups, as well as the disproportionate impacts on the physical and mental well-being of female and non-White HCWs. Importantly, this pilot study provides the scientific rationale for conducting the main study, particularly in the absence of pandemic restrictions, to answer this equity-focused and policy-relevant research question. In addition, these findings highlight the urgent need to update current respirator design and testing standards. Canadian standards and manufacturers should include anthropometric data that reflects gender and ethnic diversity to ensure that PPE provides equitable protection for all HCWs.

## Supporting information

**S1 Table. Objectives, Outcome Measures, and Analysis.**
(DOCX)

**S2 Table. Raw Data.**
(DOCX)

**S1 File. Qualitative Survey Design.**
(DOCX)

**S2 File. Study Amendment.**
(DOCX)

**S3 File. Additional Results of the Qualitative Survey.**
(DOCX)

## Acknowledgments

We would like to thank Jeff Mallany, Senior Safety Specialist (Hamilton Health Sciences), and Bonnie Peacock, Safety Fit Tester (Hamilton Health Sciences), for their expertise and willingness to support this research project.

## Author contributions

**Conceptualization:** Fatima Sheikh, Alison E. Fox-Robichaud.

**Data curation:** Fatima Sheikh.

**Formal analysis:** Fatima Sheikh.

**Funding acquisition:** Zeinab Hosseinidoust.

**Investigation:** Fatima Sheikh.

**Methodology:** Fatima Sheikh, Myrna Dolovich, Lisa Schwartz, Zeinab Hosseinidoust, Alison E. Fox-Robichaud.

**Project administration:** Fatima Sheikh.

**Supervision:** Myrna Dolovich, Lisa Schwartz, Zeinab Hosseinidoust, Alison E. Fox-Robichaud.

**Visualization:** Fatima Sheikh.

**Writing – original draft:** Fatima Sheikh.

**Writing – review & editing:** Fatima Sheikh, Myrna Dolovich, Lisa Schwartz, Sarah Khan, Zeinab Hosseinidoust, Alison E. Fox-Robichaud.

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
