## [Decision Letter · Decision Letter 0]

19 Aug 2025

Dear Dr. Fox-Robichaud,

Thank you for submitting your manuscript to PLOS ONE. After careful consideration, we feel that it has merit but does not fully meet PLOS ONE’s publication criteria as it currently stands. Therefore, we invite you to submit a revised version of the manuscript that addresses the points raised during the review process.

We look forward to receiving your revised manuscript.

Kind regards,

Christophe Curti

Academic Editor

PLOS ONE

Journal Requirements:

“This study was funded by the Canadian Institute for Health Research (CIHR) (202005VR4-447804-CCF) for one year beginning June 1st, 2020. FS was also supported by an Ontario Graduate Fellowship Award during the conduct of this study.”

4. We note that your Data Availability Statement is currently as follows: All relevant data are within the manuscript and in Supporting Information files.

Reviewers' comments:

Reviewer's Responses to Questions

**Comments to the Author**

1. Is the manuscript technically sound, and do the data support the conclusions?

Reviewer #1: Partly

Reviewer #2: Partly

2. Has the statistical analysis been performed appropriately and rigorously?

Reviewer #1: Yes

Reviewer #2: No

3. Have the authors made all data underlying the findings in their manuscript fully available?

Reviewer #1: No

Reviewer #2: No

4. Is the manuscript presented in an intelligible fashion and written in standard English?

Reviewer #1: Yes

Reviewer #2: Yes

Reviewer #1: 1. Data Policy: Plos One's data policy does not permit author contact to be the only source of the data set in most cases. This study does not, under my reading of the policy, meet any of the exceptions. The full data set should be uploaded to an open data repository.

2. Inadequate literature review. Other researchers have conducted qualitative and quantitative investigations into the impact of ethnicity and gender on respirator fit. These works are a) not mentioned in this paper and b) the originality / importance of this data are not established in relevance to prior work. While authors are correct sex has been more extensively studied than ethnicity, they are incorrect that ethnicity has not been much studied (lines 318-319). I would recommend the authors review the following literature on both gender and ethnicity:

--- Prospective observational study of gender and ethnicity biases in respiratory protective equipment for healthcare workers in the COVID-19 pandemic by C. Carvalho et al 2021

--- The influence of gender and ethnicity on facemasks and respiratory protective equipment fit: a systematic review and meta-analysis by J Chopra et al 2021

--- The effect of N95 designs on respirator fit and its associations with gender and facial dimensions by Hasni et al 2021

--- Respirator Fit and Facial Dimensions of Two Minority Groups by W Brazile et al 1998

--- The Effect of Subject Characteristics and Respirator Features on Respirator Fit by Z Zhuang et al 2005

--- The Effect of Gender and Respirator Brand on the Association of Respirator Fit with Facial Dimensions by R. Oestenstad et al 2007

--- P2/N95 filtering facepiece respirators: Results of a large-scale quantitative mask fit testing program in Australian health care workers by M Milosevic et all 2021

--- 316 Fit of N95 filtering facepiece respirators influenced by gender, design of facepiece, and activity engaged in use by Glass et all 2013

--- The PPE Pandemic: Sex-Related Discrepancies of N95 Mask Fit by L Christopher 2021

--- Evaluation of a Large-Scale Quantitative Respirator-Fit Testing Program for Healthcare Workers: Survey Results by Wilkinson et al 2010

3. Although this study was unable to obtain sufficient participants for statistical significance, I do believe the preliminary quantitative fit test findings are interesting and may potentially be of benefit to the scientific community. I am not sure that they warrant an original full paper - but if there is a brief report section, this would be a perfect fit. The significance of this paper, and it’s publish-ability, will depend partly on what it contributes in relation to existing literature (see point 2).

4. I would recommend the authors be more upfront / explicit about the inadequate sample size in the abstract and introduction. There is repeated mention of the study not meeting feasibility criteria. I would suggest changing the language to explicitly discuss the small sample size and reasons for this limitation. I believe the "feasibility" language currently used hides the utility of the study. Also, the current language is somewhat misleading - as this isn't a study to see if a method is feasible, as the method is well established in both scientific and industrial use.

5. The analysis of qualitative survey findings is lacking. If the qualitative findings are not robust enough to be useful, I would suggest leaving them out of the paper and instead making them available in the dataset for interested researchers. Otherwise, discussion and analysis are needed.

Reviewer #2: I have reviewed the above manuscript, which addresses an important and timely question about the fit, comfort, and breathability of N95 respirators among healthcare workers of diverse genders and ethnicities. The work is relevant to occupational health, equity in PPE design, and pandemic preparedness. The mixed-methods approach is appropriate and offers valuable context to the quantitative fit test results.

However, several areas would benefit from clarification and additional detail before the manuscript can be considered for publication. My major concerns relate to:

Comment 1: The integration of qualitative and quantitative results, which is mentioned but not described in enough methodological detail.

Comment 2: The aggregation of all non-White participants into one group, which may obscure important differences.

Comment 3: Limited discussion of the implications of the feasibility shortfall, both methodologically and in terms of generalizability.

Comment 4: Greater contextualization of the “out-of-range” anthropometric finding from Figure 2.

Below, I provide detailed comments, embedded within the manuscript text for ease of reference.

Sincerely,

Anahita Fakherpour

Embedded Reviewer Comments in Manuscript

Abstract (lines 23–48)

Comment 1: Clarify why the feasibility target was set at 100 participants, and what the reduced sample size means for interpreting both quantitative and qualitative results. Consider adding a sentence on whether this impacts the power to detect differences.

Background (lines 62–96)

Comment 2: The background section is strong, but the reliance on pre-pandemic anthropometric data references (e.g., 2005) should be supplemented with more recent post-COVID PPE design literature.

Comment 3: If any Canadian-specific anthropometric studies exist, briefly mention them to strengthen the rationale for conducting this work in Canada.

Methods (lines 106–198)

Comment 4: More detail is needed on how the “statistics-by-theme joint display” was created—did integration involve iterative coding, and how were disagreements resolved?

Comment 5: Justify the requirement that 50% of participants be non-White or have other specific characteristics. Was this based on effect size estimation, diversity representation targets, or prior literature?

Comment 6: Explain why no exploratory non-parametric statistics (e.g., Mann–Whitney U tests) were used, as these can be informative even in small samples.

Comment 7: Indicate whether fit testers were blinded to participant ethnicity and gender to reduce potential observer bias.

Comment 8: The exclusion criteria should be reported precisely.

Results – Feasibility (lines 200–231)

Comment 9: The feasibility shortfall is described, but more interpretation is needed. Were these challenges purely pandemic-related, or do they indicate systemic issues in recruiting HCWs for PPE research?

Comment 10: Consider quantifying the recruitment rate per week before and after the study amendment to better illustrate the impact of additional sites.

Results – Quantitative Fit Test (lines 233–259)

Comment 11: For anthropometric results, consider providing percentage differences to make the magnitude of differences more interpretable.

Comment 12: The finding that 75% of participants fall outside the NIOSH bivariate panel range is important—state briefly in Results what this implies for respirator design testing standards.

Results – Qualitative (lines 261–297)

Comment 13: Tables 4a and 4b should include scale anchors (e.g., 0 = poor, 4 = excellent) directly in the table for clarity.

Comment 14: In reporting comfort and breathability scores, you may wish to highlight how these subjective measures aligned—or conflicted—with quantitative fit factors.

Discussion (lines 299–360)

Comment 15: The comparison to prior literature is solid, but the discussion should explicitly address whether the trends observed (e.g., gender/ethnic disparities in fit) are likely to persist outside pandemic surge conditions.

Comment 16: Acknowledge the absence of participants with religious head coverings or facial hair, as this limits the generalizability to all HCW subgroups.

Conclusion (lines 362–368)

Comment 17: Strengthen the policy recommendations by explicitly calling for updates to the NIOSH bivariate panels and Canadian standards to incorporate gender and ethnic diversity in respirator design, along with specific recommendations for manufacturers.

Figures and Tables

Comment 18: Include sample sizes (n) in figure legends for transparency.

Comment 19: In Figure 2, annotate the proportion of participants outside the panel range for emphasis.

**Do you want your identity to be public for this peer review?** For information about this choice, including consent withdrawal, please see our Privacy Policy

Reviewer #1: **Yes: ** Eugenia O'Kelly

Reviewer #2: No

---

## [Author Response · Author response to Decision Letter 1]

3 Oct 2025

JOURNAL REQUIREMENTS

Response: Thank you. We have re-formatted our manuscript to align with PLOS One’s style requirements.

Please note that funding information should not appear in any section or other areas of your manuscript. We will only publish funding information present in the Funding Statement section of the online submission form. Please remove any funding-related text from the manuscript.

Response: We have removed all funding information from the manuscript.

Thank you for stating the following financial disclosure:

“This study was funded by the Canadian Institute for Health Research (CIHR) (202005VR4-447804-CCF) for one year beginning June 1st, 2020. FS was also supported by an Ontario Graduate Fellowship Award during the conduct of this study.”

Response: We have added a statement indicating the funder had no role in any of the study proceedings and included a revised statement in our cover letter.

4. We note that your Data Availability Statement is currently as follows: All relevant data are within the manuscript and in Supporting Information files.

If there are ethical or legal restrictions on sharing a de-identified data set, please explain them in detail (e.g., data contain potentially sensitive information, data are owned by a third-party organization, etc.) and who has imposed them (e.g., an ethics committee).

Please also provide contact information for a data access committee, ethics committee, or other institutional body to which data requests may be sent. If data are owned by a third party, please indicate how others may request data access.

Response: Thank you. We have included a minimal data set as part of our re-submission.

Response: We have removed our ethics statement from the declarations section. It is now only mentioned under methods.

Please include captions for your Supporting Information files at the end of your manuscript, and update any in-text citations to match accordingly. Please see our Supporting Information guidelines for more information: http://journals.plos.org/plosone/s/supporting-information.

Response: Thank you. We have added captions/titles for our supporting information following the references.

Response: Thank you. We have included suggested references only where appropriate.

REVIEWER 1

Data Policy: Plos One's data policy does not permit author contact to be the only source of the data set in most cases. This study does not, under my reading of the policy, meet any of the exceptions. The full data set should be uploaded to an open data repository.

Response: Thank you for the careful review of policies. We have uploaded the de-identified raw quantitative data as part of our revisions.

Inadequate literature review. Other researchers have conducted qualitative and quantitative investigations into the impact of ethnicity and gender on respirator fit. These works are a) not mentioned in this paper and b) the originality / importance of this data are not established in relevance to prior work. While authors are correct sex has been more extensively studied than ethnicity, they are incorrect that ethnicity has not been much studied (lines 318-319). I would recommend the authors review the following literature on both gender and ethnicity:

--- Prospective observational study of gender and ethnicity biases in respiratory protective equipment for healthcare workers in the COVID-19 pandemic by C. Carvalho et al 2021

--- The influence of gender and ethnicity on facemasks and respiratory protective equipment fit: a systematic review and meta-analysis by J Chopra et al 2021

--- The effect of N95 designs on respirator fit and its associations with gender and facial dimensions by Hasni et al 2021

--- Respirator Fit and Facial Dimensions of Two Minority Groups by W Brazile et al 1998

--- The Effect of Subject Characteristics and Respirator Features on Respirator Fit by Z Zhuang et al 2005

--- The Effect of Gender and Respirator Brand on the Association of Respirator Fit with Facial Dimensions by R. Oestenstad et al 2007

--- P2/N95 filtering facepiece respirators: Results of a large-scale quantitative mask fit testing program in Australian health care workers by M Milosevic et all 2021

--- 316 Fit of N95 filtering facepiece respirators influenced by gender, design of facepiece, and activity engaged in use by Glass et all 2013

--- The PPE Pandemic: Sex-Related Discrepancies of N95 Mask Fit by L Christopher 2021

--- Evaluation of a Large-Scale Quantitative Respirator-Fit Testing Program for Healthcare Workers: Survey Results by Wilkinson et al 2010

Response: We thank the reviewer for this important observation and have revised the discussion to acknowledge prior work, not already cited, examining the influence of gender and ethnicity on respirator fit. Several studies report higher fit test failure rates among women and ethnic minority groups, particularly Asian healthcare workers, while others attribute fit primarily to individual facial dimensions and respirator design. These mixed findings underscore the importance of conducting population-specific studies, as respirator fit reflects both individual and group-level anthropometric variation. In light of Canada’s increasingly diverse healthcare workforce, we argue that locally derived evidence is essential to ensure respirator design and fit testing protocols adequately meet the needs of all healthcare workers.

Although this study was unable to obtain sufficient participants for statistical significance, I do believe the preliminary quantitative fit test findings are interesting and may potentially be of benefit to the scientific community. I am not sure that they warrant an original full paper - but if there is a brief report section, this would be a perfect fit. The significance of this paper, and it’s publishability, will depend partly on what it contributes in relation to existing literature (see point 2).

Response: We appreciate the reviewer’s comment and acknowledge the limitation of our sample size. However, we believe the preliminary findings remain valuable, as they provide early evidence on respirator fit in relation to gender and ethnicity within a Canadian healthcare workforce. As outlined in our revised discussion, this work contributes to the limited literature on population-specific respirator fit and underscores the importance of generating locally relevant data. To our knowledge, none of the existing studies on gender and ethnicity and respirator fit have been conducted in Canada, where the healthcare workforce is increasingly diverse; thus, locally derived evidence is essential to inform respirator design and fit testing protocols.

I would recommend the authors be more upfront / explicit about the inadequate sample size in the abstract and introduction. There is repeated mention of the study not meeting feasibility criteria. I would suggest changing the language to explicitly discuss the small sample size and reasons for this limitation. I believe the "feasibility" language currently used hides the utility of the study. Also, the current language is somewhat misleading - as this isn't a study to see if a method is feasible, as the method is well established in both scientific and industrial use.

Response: Thank you for this note. We have revised our study abstract and conclusion to explicitly state the limitations related to the sample size. Below is the revised text:

From the Abstract: “This study was limited by a small sample size, as COVID-19 pandemic restrictions prevented adequate recruitment to detect differences between groups.”

From the Conclusion: “This study was conducted during the COVID-19 pandemic, when restrictions on in-person research limited recruitment and resulted in a small sample size. While this precluded statistical comparisons between groups, the preliminary findings provide important insights into gender- and ethnicity-related variation in respirator fit among Canadian healthcare workers.”

The analysis of qualitative survey findings is lacking. If the qualitative findings are not robust enough to be useful, I would suggest leaving them out of the paper and instead making them available in the dataset for interested researchers. Otherwise, discussion and analysis are needed.

Response: Thank you for this note. Our intention in including the qualitative survey was to describe the experiences of healthcare workers with respirator fit, comfort, and impact on well-being. We agree that the qualitative data are descriptive and preliminary and have revised the manuscript to make this purpose explicit. We have expanded the discussion of these findings to highlight how they complement the quantitative results and underscore the need for further in-depth qualitative research. We believe retaining these data adds value by situating the fit test results within the lived experiences of HCWs, particularly as issues of comfort and psychological burden are central to understanding respirator use.

REVIEWER 2

I have reviewed the above manuscript, which addresses an important and timely question about the fit, comfort, and breathability of N95 respirators among healthcare workers of diverse genders and ethnicities. The work is relevant to occupational health, equity in PPE design, and pandemic preparedness. The mixed-methods approach is appropriate and offers valuable context to the quantitative fit test results. However, several areas would benefit from clarification and additional detail before the manuscript can be considered for publication.

Response: Thank you for taking the time to review our manuscript and provide detailed feedback. We sincerely appreciate the opportunity to revise and improve our manuscript.

Abstract (lines 23–48)

Clarify why the feasibility target was set at 100 participants, and what the reduced sample size means for interpreting both quantitative and qualitative results. Consider adding a sentence on whether this impacts the power to detect differences.

Response: Thank you for this comment. We have revised our abstract to explicitly state the small sample size and insufficient power to detect statistically significant differences between groups and have added the rationale for “100 participants” to Table 1.

Background (lines 62–96)

The background section is strong, but the reliance on pre-pandemic anthropometric data references (e.g., 2005) should be supplemented with more recent post-COVID PPE design literature.

Response: Thank you for the kind words! We have included more recent references in our introduction (lines 100-102) . We have also included a more detailed analysis of these studies in relation to our work in the discussion (lines 437-447).

If any Canadian-specific anthropometric studies exist, briefly mention them to strengthen the rationale for conducting this work in Canada.

Response: Based on comments from the other review, we have made modifications in our introduction (lines 100-102) and in our discussion (lines 437-447) to highlight the existing body of work on this topic, the lack of Canadian data, and how our study fills an important gap.

Methods (lines 106–198)

More detail is needed on how the “statistics-by-theme joint display” was created—did integration involve iterative coding, and how were disagreements resolved?

Response: We thank the reviewer for this suggestion. We have added further detail in the Methods to clarify how the “statistics-by-theme joint display” was created. Specifically, integration of the quantitative and qualitative findings was conducted through an iterative process in which authors independently reviewed the descriptive survey responses and then collaboratively organized the findings into a joint display to highlight points of convergence and divergence. Disagreements were resolved through discussion until consensus was reached. This approach ensured that both data sources were represented transparently and that the joint display accurately reflected the descriptive nature of the qualitative findings alongside the quantitative results. These changes are reflected in lines 207-212.

Justify the requirement that 50% of participants be non-White or have other specific characteristics. Was this based on effect size estimation, diversity representation targets, or prior literature?

Response: We appreciate the reviewer’s request for clarification. The requirement that at least 50% of participants be non-White was not based on effect size estimation but was instead established a priori to ensure adequate representation of participants from diverse ethnic groups. Prior literature has demonstrated that respirator fit may vary across ethnic groups, with Asian and other non-White healthcare workers experiencing higher fit failure rates compared to White workers. Without intentional recruitment, there is a risk that studies conducted in Canada would overrepresent White participants, limiting the ability to explore these equity-relevant differences. Our sampling requirement was therefore designed to align with diversity representation targets, to generate more inclusive and locally relevant evidence. We have clarified this in lines 177-181.

Explain why no exploratory non-parametric statistics (e.g., Mann–Whitney U tests) were used, as these can be informative even in small samples.

Response: We thank the reviewer for this suggestion. We considered the use of exploratory non-parametric tests such as Mann–Whitney U; however, given the small sample size even non-parametric analyses would have had extremely limited interpretability and risked overstating differences. Instead, we chose to present descriptive statistics and qualitative findings in parallel, which we believe more appropriately reflect the preliminary and exploratory nature of the study. This is described in lines 200-202.

Indicat

---

## [Decision Letter · Decision Letter 1]

22 Oct 2025

Evaluating N95 Respirator Designs: A Mixed-Methods Pilot and Feasibility Study

PONE-D-25-34031R1

Dear Dr Fox-Robichaud, we are pleased to inform you that your manuscript has been judged scientifically suitable for publication and will be formally accepted for publication once it meets all outstanding technical requirements.

Kind regards,

Christophe Curti

Academic Editor

PLOS ONE

Additional Editor Comments (optional):

Reviewers' comments:

Reviewer's Responses to Questions

**Comments to the Author**

Reviewer #1: All comments have been addressed

Reviewer #2: (No Response)

2. Is the manuscript technically sound, and do the data support the conclusions?

Reviewer #1: Yes

Reviewer #2: Yes

3. Has the statistical analysis been performed appropriately and rigorously?

Reviewer #1: Yes

Reviewer #2: Yes

4. Have the authors made all data underlying the findings in their manuscript fully available?

Reviewer #1: Yes

Reviewer #2: Yes

5. Is the manuscript presented in an intelligible fashion and written in standard English?

Reviewer #1: Yes

Reviewer #2: Yes

Reviewer #1: Thank you for your edits. I believe all my concerns have been adequately addressed.

Supporting information and data is clear and helpful.

I think the inclusion of survey qualitative findings greatly enhances the study.

Reviewer #2: (No Response)

**Do you want your identity to be public for this peer review?** For information about this choice, including consent withdrawal, please see our Privacy Policy

Reviewer #1: **Yes: ** Eugenia O'Kelly

Reviewer #2: No

---

## [Editor Report · Acceptance letter]

PONE-D-25-34031R1

PLOS ONE

Dear Dr. Fox-Robichaud,

I'm pleased to inform you that your manuscript has been deemed suitable for publication in PLOS ONE. Congratulations! Your manuscript is now being handed over to our production team.

Kind regards,

on behalf of

Professor Christophe Curti

Academic Editor

PLOS ONE